

# Adjusting Diurnal Error in In-Situ Soil Moisture Measurements via Fourier Time-Filtering Using Land Surface Model Datasets

Junnyeong Han[1], Eunkyo Seo[1,2], Paul A. Dirmeyer[2]

[1] Department of Environmental Atmospheric Sciences, Pukyong National University, Busan, 48513, Republic of Korea
[2] Center for Ocean-Land-Atmosphere Studies, George Mason University, Fairfax, Virginia, 22030, United States

*Correspondence to*: Eunkyo Seo (eseo@pknu.ac.kr)

**Abstract.** Soil moisture (SM) measurements obtained via dielectric-based sensors are widely used in hydrological and climate studies. However, these measurements exhibit significant temperature sensitivity due to the Maxwell–Wagner polarization effect, causing an unrealistic diurnal cycle having spurious daytime peaks. This study introduces a Fourier
transform-based method to correct such temperature-induced errors using physically consistent diurnal patterns from land surface model (LSM) reanalysis datasets (ERA5-Land and MERRA-2). The proposed approach adjusts the spectral power of the SM diurnal cycle to align with model-derived patterns constrained by conservation of mass, resulting in physically realistic SM behavior—peaking in the morning and decreasing during the daytime due to evapotranspiration. Validation against non-dielectric reference sensors indicates that the adjusted SM measurements are significantly improved. The diurnal correlation
between SM and soil temperature shifts from predominantly positive to negative, particularly evident in regions with large diurnal temperature ranges and dry climates. Furthermore, applying this method to flux tower observations improves the characterization of land–atmosphere interactions by depicting the energy-limited process at sub-daily timescales, where increased incoming radiation during the daytime drives enhanced latent heat flux and subsequently reduces SM. Overall, this Fourier transform-based adjustment enhances the verity of in-situ soil moisture observations, promoting accurate sub-daily
analyses of soil moisture dynamics and enabling improved understanding of land–atmosphere coupling processes.

## 1 Introduction

Soil moisture (SM) is a critical land variable, playing a key role in hydrological and meteorological processes through land–atmosphere (L–A) interactions (Seneviratne et al., 2010). It is also an essential climate variable with a memory up to 1–
2 months (Seo and Dirmeyer, 2022a; Rahmati et al., 2024), thereby contributing to predictability at the subseasonal timescale (Koster et al., 2011; Seo et al., 2019). SM strongly influences precipitation, energy partitioning, and the diurnal cycles of key land variables—including evapotranspiration (ET), temperature, humidity, latent heat flux (LH), and sensible heat flux (SH)— by controlling ET, which supplies moisture to the atmosphere during the daytime and influences boundary layer characteristics depending on surface moisture availability and energy constraints (Betts, 2003; Seo and Dirmeyer, 2022b; Dai et al., 1999).



These SM-driven surface and atmospheric processes form a fundamental component of L-A coupling, by which SM modulates
the partitioning of land surface fluxes, affects planetary boundary layer (PBL) development and entrainment, and ultimately
influences cloud formation and precipitation (Santanello et al., 2018). In energy-limited regions, where ET is primarily
constrained by incoming solar radiation, sufficient SM availability leads to relatively enhanced ET (Hsu et al., 2024; Dong et
al., 2022). For instance, higher ET is particularly dominant in regions with high humidity and a shallow boundary layer, as

moist air supports plant transpiration. In contrast, in water-limited regions, where SM availability is low, SM plays a key role
in determining the partitioning between SH and LH to the atmosphere (Seo et al., 2024).

Although diurnal SM variability accounts for 1–2% of its total variance in observations (Fig.1, bottom left), L-A
interactions are predominantly active at sub-daily timescales during daytime, primarily driven by surface fluxes resulting from
incoming solar radiation (Seo and Dirmeyer, 2022b; Yin et al., 2023). The investigation of L-A interactions has relied on low-

frequency variation data (e.g., daily, monthly, and yearly), which limits our understanding of the actual coupling mechanisms
throughout the diurnal cycle (Findell et al., 2024), especially how moisture stress limits ET during the warmest part of the day.
Daily averaged land variables, which include nighttime conditions, tend to suppress daytime-dominant interactions, leading to
an underestimation of coupling strength driven by surface fluxes (Yin et al., 2023). Seo and Dirmeyer (2022b) demonstrated
that the diurnal coevolution of water and thermal energy budgets within atmospheric boundary layer in terms of L–A

interactions exhibit an asymmetric structure in that phase-space, with a strong diurnal cycle of heat content but a clear semi-
diurnal cycle for moisture, driven by the interplay of changing surface evaporation and the depth of the boundary layer.
However, daily mean values are not enveloped within the closed diurnal cycle in the phase-space of water and heat budgets,
as the computed mean state typically does not correspond to any real conditions experienced during a full 24-hour cycle.
Therefore, sub-daily analyses—particularly those capturing the full diurnal evolution of surface fluxes and boundary layer

development—are essential for accurately characterizing the phase and intensity of L-A interactions. Despite their importance,
studies of L-A interactions at sub-daily timescales using in-situ observations remain limited, highlighting the need for
observational datasets with fine temporal resolution.

The International Soil Moisture Network (ISMN; Dorigo et al., 2021), which provides hourly SM time series, has been
widely used to investigate SM behavior on a range of timescales. Methods for SM measurement can be categorized into direct

and indirect approaches. The gravimetric method, a direct approach, is the most accurate but is destructive (disturbs the soil
being measured), time-consuming, and labor-intensive, making it unsuitable for real-time or long-term climate monitoring.
Given these limitations, many indirect methods, including cosmic ray sensors, neutron probes, and sensors based on time
domain transmissometry (TDT), time domain reflectometry (TDR), frequency domain reflectometry (FDR), electric
capacitance, and impedance, have been widely used. However, since these methods estimate SM based on water's effect on

various electromagnetic properties of matter, they introduce inherent errors, and the sensors themselves have limitations. For
instance, cosmic ray sensors and neutron probes offer accurate SM estimates with large- and point-scale footprints, respectively,
but they are expensive and come with significant limitations. Cosmic ray sensors require regular maintenance, their
measurement depth varies with soil water content, and they have strong installation constraints (Zreda et al., 2012). Neutron





probes rely on radioactive materials, posing safety risks and requiring strict regulatory compliance (Hunduma and Kebede, 2020; Sharma et al., 2018; Lekshmi et al., 2014). On the other hand, other indirect methods (e.g., TDT, TDR, FDR, capacitance, and impedance sensors) are widely used due to their low cost, ease of operation, relatively simple installation and management, fast response, automation and multiplexing capability, non-destructive measurement, and safety from radiation hazards. These sensors estimate SM based on the dielectric constant, which makes them highly sensitive to temperature variations (Mane et al., 2024; Francesca et al., 2010; Mittelbach et al., 2012; Ojo et al., 2015; Rowlandson et al.,2013).

Temperature sensitivity arises from Maxwell-Wagner polarization, a phenomenon occurring at the interface of contrasting matter phases due to differences in dielectric permittivity and electrical conductivity. This interfacial polarization alters the effective bulk dielectric permittivity (Chelidze and Gueguen, 1999; Chen and Or, 2006a). In porous materials such as soil, the interfaces between water, air, and mineral particles have different electrical properties. Temperature variations influence the separation, mobility, and relaxation time of electrical charges, thereby altering the bulk dielectric response measured by the sensor. The temperature sensitivity of Maxwell–Wagner polarization depends on frequency: at low frequencies (e.g., <100 MHz), the response is dominated by interfacial polarization, which increases with temperature due to enhanced ion mobility and electrical conductivity (Chen and Or, 2006b). Low-frequency sensors (e.g. capacitance, impedance, and FDR) are widely used due to due to their low cost, ease of installation, and suitability for automated monitoring systems. However, these sensors often exhibit temperature-dependent artifacts, resulting in spurious daytime SM peaks and a positive temporal correlation with the diurnal cycle of temperature, even though the reality of diurnal SM minimum near the surface is observed in the daytime (Kosa, 2009). Such spurious diurnal SM evolution, in the absence of precipitation, contradicts the water balance at land surface.

Previous studies have attempted to correct for the temperature sensitivity in indirect SM measurements. These approaches can be classified into mechanistic methods, which aim to formulate corrections based on the physical mechanisms underlying temperature effects on dielectric properties, and empirical correction techniques informed by the statistical properties of observational data. For mechanistic approaches, the temperature sensitivity of soil dielectric permittivity has been investigated through its frequency-dependent response, with a liner model of the temperature dependence of the real part of permittivity serving as a basis for temperature correction in SM estimation (Skierucha et al., 2024). A moisture deviation coefficient has also been proposed to quantify temperature-induced biases, in which permittivity changes lead to systematic measurement errors. To mitigate these, laboratory calibration, energy and water transfer modeling, and machine learning techniques have been suggested (Wilczek et al., 2023). For empirical correction approaches, Chanzy et al. (2012) suggested a daily correction coefficient calculated from diurnal variations in permittivity and temperature, potentially influenced by SM and conductivity. It was applied to adjust measured permittivity, using time periods with minimal moisture change (i.e., early morning or late afternoon) for stable estimation.

While these approaches offer useful corrections, most rely on site-specific assumptions or sensor-dependent characteristics that are not easily extrapolated, highlighting the need for a more generally applicable correction method. Therefore, this study proposes a type of empirical correction method using Fourier transform to better represent the realism of the SM diurnal cycle in accordance with the land surface water balance. To implement this correction, we utilize land surface reanalysis datasets





generated by state-of-the-art physical models, which ensure water budget closure, particularly capturing the inverse relationship between SM and ET. Based on the diurnally adjusted SM observations, we further assess the impact of the time-filtering method across different land cover types and background climatological conditions, as well as its influence on land-atmosphere interactions, using flux tower observations, coupled with LH at sub-daily timescales.

## 2 Data

### 2.1 In-situ observation

The International Soil Moisture Network (ISMN; Dorigo et al., 2021) was initiated to provide a standardized data system that integrates data from various in-situ monitoring networks through quality control and harmonization. SM observations had been collected and distributed in disparate formats, making it difficult to incorporate them into research. ISMN hosts data from 77 networks with more than 2,900 stations distributed globally, though majority are concentrated in the Continental United States (CONUS) and Europe. It provides SM and soil temperature (TS) observations at various depths, along with some precipitation and air temperature data. However, different measurement sensors are used at each station, even for the same variable, such that SM data comes from a diverse set of sensors, including impedance, capacitance, cosmic-ray, TDR, TDT, and FDR methods. Most are provided at an hourly temporal resolution in UTC, with a consistent format and unit convention. ISMN also provides various quality flags for identifying measurements that are unrealistic, scientifically implausible or statistical outliers, and offers metadata describing station characteristics, including soil texture, climate classification (i.e., Köppen-Geiger), land cover type, and geolocation information. Importantly, the sensors' manufacturer and model are identified for every time series. The ISMN dataset has been widely used to understand the nature of soil physics and evaluate satellite-based and modeled SM products (e.g., Gruber et al., 2020; Beck et al., 2021; Seo et al., 2021; Yi et al., 2023).

In this study, soil moisture and temperature measurements within the top 10 cm of soil are used, encompassing all available shallow depth measurements. Since in-situ observations often contain missing or unreliable measurements, we filter the data to include only those with a quality flag marked as 'G (Good)' and utilize only days with complete 24-hour observations, as our focus is on the entire diurnal cycle. For obtaining enough diurnal samples, stations with at least 92 days of overlapping data for both SM and temperature are used; 1,058 stations are ultimately used across the globe (Fig. 1). Among these, only 10 stations use non-dielectric-based sensors; all other sites measure SM with dielectric permittivity-based sensors (hereafter, dielectric-based sensors). The dielectric-based sensors estimate SM based on the dielectric permittivity of soil, which is inherently sensitive to temperature variations. This temperature dependence introduces diurnal errors in the measurements, which appear as a physical water imbalance such as a spurious positive correlation between soil moisture and temperature. However, some dielectric-based sensors that operate at high frequencies (e.g. TDT and TDR) exhibit a limited sensitivity to temperature, showing a realistic physical relationship between soil moisture and temperature with a negative correlation—similar to non-dielectric-based cosmic-ray sensors (Fig. 2a). Thus, 100 stations (10: non-dielectric-based sensors, 90: high-frequency dielectric-based sensors) serve as reference sites to validate the SM diurnal cycle of in-situ measurements.



**2.2 Land Reanalysis Datasets**

The European Centre for Medium-Range Weather Forecasts (ECMWF) Reanalysis version 5 Land (ERA5-Land) (Muñoz-Sabater et al., 2021) is a global reanalysis dataset. It employs the Carbon Hydrology-Tiled ECMWF Scheme for Surface Exchanges over Land (CHTESSEL) as its land surface model. Unlike ERA5 (Hersbach et al., 2020), which utilizes a simplified extended Kalman filter (SEKF) that assimilates European Remote Sensing Satellite (ERS)-1/-2 and Advanced Scatterometer

(ASCAT), ERA5-Land is an offline simulation solely driven by ERA5 atmospheric forcing for near-surface meteorological variables. ERA5-Land provides hourly outputs at a spatial resolution of 9 km (0.1°) from 1950 to the present, compared to ERA5 reanalysis with 31 km resolution. To account for topographic discrepancies between the ERA5 and ERA5-Land grids, air temperature is corrected using a lapse rate derived from ERA5 lower-tropospheric temperature profiles. Subsequently, surface pressure and specific humidity are recalculated based on the adjusted temperature and elevation, assuming constant

relative humidity. The enhanced spatial resolution and refined topographic representation contribute to the improvement in land surface estimates. While most differences from ERA5 are methodological and related to spatial resolution enhancement, ERA5-Land also includes physical updates such as revised soil thermal conductivity in frozen conditions, improved soil water balance closure, and an explicit treatment of rainfall over snow. This study uses ERA5-Land because of its superior performance in reproducing realistic SM time series against in-situ measurements across the globe (see Fig. 3a in Beck et al.,

2021).

The Modern-Era Retrospective analysis for Research and Applications, Version 2 (MERRA-2; Gelaro et al., 2017), is another global reanalysis dataset, developed by the National Aeronautics and Space Administration (NASA) Global Modeling and Assimilation Office (GMAO). It is produced using the Goddard Earth Observing System version 5 (GEOS5) atmospheric data assimilation system and Gridpoint Statistical Interpolation (GSI) (Wu et al., 2002; Kleist et al., 2009) analysis scheme,

which performs grid-based analysis using a 3D-Var approach. MERRA-2 uses the Catchment LSM, with offline LSM runs driven by atmospheric conditions from the GEOS-5 system operating in "replay" mode, during which modelled precipitation is corrected using satellite- and gauge-based datasets, resulting in a more realistic representation of SM. This catchment-based approach realistically captures topographical influences of hillslope hydrology and improves the simulation of land surface processes such as runoff and evaporation by explicitly partitioning each grid cell into saturated, unsaturated, and wilting regions

that exhibit distinct hydro-meteorological behaviors (Koster et al., 2000). MERRA-2 provides hourly SM estimates for the surface layer (0–5 cm) and root-zone (0–100 cm) from 1980 to the present, with a spatial resolution of 0.625° × 0.5°.

The Global Land Data Assimilation System (GLDAS; Rodell et al., 2004) product is another candidate for incorporating physically constrained land surface products driven by near-surface atmospheric forcing to adjust the diurnal cycle in SM measurements. Although GLDAS version 2.1 exhibits diurnal behavior similar to that of reference observations, it is not

employed in this study due to its 3-hourly temporal resolution, which limits its ability to capture the 24-hour diurnal cycle via the application of a Fourier transform-based adjustment used in this study. This study uses LSM-derived SM and temperature in the surface layer (0–7 cm for ERA5-Land and 0–5 cm for MERRA-2) for diurnal adjustment, due to their physically





constrained, realistic behavior (Seo and Dirmeyer, 2022a). For comparison with the observations, the grid cell nearest to each observation site is selected from the respective reanalysis datasets.

### 2.3 Flux tower observations

The ISMN dataset does not include measurements of land surface heat fluxes, so this study employs flux tower observations to investigate the effect of adjusting the diurnal cycle of SM on the representation of observed L-A coupling at sub-daily timescales. In flux tower observations, land surface heat fluxes are measured using eddy covariance methods (Pastorello et al., 2020). SM at these sites is measured using multiple types of dielectric-based sensors (e.g. TDR, FDR, etc.) (Zhang and Yuan, 2020; Op de Beeck et al., 2018), which suffer from the same temperature sensitivity errors described previously. This can lead to a spurious SM peak during the daytime, despite high ET, even if precipitation is absent.

FLUXNET2015, the latest major release of globally harmonized flux tower observations, provides Tier 1 data accompanied by quality flags for each variable, along with the flag of uncertainty and gap-filled data. This network provides SM measurements at depths up to 100 cm from the mid-1990s up to 2015. AmeriFlux also provides flux observations through the present day across a wide range of ecosystem and land cover types, primarily across North and South America (Novick et al., 2018), where SM is generally reported at two layers: a top layer (typically up to 10 cm) and a bottom layer (up to 100cm) (Qiu et al., 2016). The land surface variables are harmonized using the global Flux Processing Standard, which standardizes metadata (e.g., variable name, units and formats) at hourly or half-hourly resolution (Chu et al., 2023). Additionally, the Integrated Carbon Observation System (ICOS; Heiskanen et al., 2022) is a European research infrastructure established to provide standardized, long-term, high-precision observations of greenhouse gas (GHG) concentrations and fluxes across the atmosphere, terrestrial ecosystems, and oceans. In ICOS, dielectric-based sensors measure SM at depths of 5, 10, 20, 50 and 100 cm (Op de Beeck et al., 2018). This study further uses the warm-winter-2020 and drought-2018 datasets from ICOS, which were extended and expanded from FLUXNET2015 to investigate anomalously warm winter of 2020 and the extreme drought in Europe in 2018, respectively.

This study uses available flux tower observations spanning the past 30 years (FLUXNET2015: mid-1990s–2015; AmeriFlux: mid-1990s–present; ICOS: 2008–present), incorporating SM measurements at depths up to 10 cm, which ensures consistency across all sites. Where FLUXNET2015 spatially and temporally overlaps the AmeriFlux and ICOS data, the FLUXNET2015 is given priority, and the other datasets are used to extend the temporal coverage of the FLUXNET2015 data. Among the 342 stations with both SM and LH observations, 178 stations with at least 92 days of complete 24-hour records (as defined in the ISMN dataset) are selected for this study (Fig. S1). Using SM and LH data from these 178 stations, the analysis is conducted across multiple temporal scales, including diurnally adjusted hourly, original hourly, daily and monthly averaged SM data. The daily and monthly timescales are derived by averaging the original hourly time series. The flux tower sites are mostly located in mid-latitude regions, where vegetated environments typically exhibit wet soil, representing energy-limited regimes consistent with the ecological monitoring objectives of the networks. In these regions, sufficient daytime energy availability enhances ET, which subsequently leads to drying out SM.



## 3 Methodology

### 3.1 Data preprocessing

To examine the diurnal variation of SM primarily driven by ET, which is a sink term in the water budget at the land surface, days when the precipitation source term is present have been excluded from all three datasets (in-situ measurements, LSM outputs, and flux tower observations). This is because incoming shortwave radiation during the daytime enhances ET, leading to a decrease in SM, a signal that can be obscured by precipitation. Rainy days are filtered out using a standard deviation-based approach, instead of incorporating ground-based precipitation observations, due to their limited availability in ISMN, incompleteness, and spatial inconsistency with other variables (e.g., SM and ET) (Seo and Dirmeyer, 2022b). To exclude rainy days, hourly SM measurements are aggregated to a daily resolution to estimate the day-to-day SM tendency. When the tendency exceeds a threshold of +1.5 standard deviations, calculated over the entire analysis period, both the current and previous days are identified as rainy days and excluded from the adjustment (Step 1 in Fig. 4a). This threshold value is adopted because it best corresponds to days with observed precipitation in available in-situ data. For the LSMs, rainy days are excluded based on their included precipitation data, using a threshold of daily total precipitation exceeding 0.1mm.

Additionally, low-frequency variability in SM is filtered out to isolate its diurnal component by subtracting a 24-hour centered moving average from the original hourly time series, resulting in the hourly anomalies used in the diurnal adjustment (Step 2 in Figs. 4b, c). In the calculation of the running mean over a 24-hour window, data gaps or the exclusion of rainy days may result in windows containing fewer than 24 observations. In such cases, the mean is still computed using the available data points within each window, provided that at least one valid observation is present.

### 3.2 Diurnal Soil Moisture adjustment

To adjust the diurnal cycle of SM, this study adopts the Fast Fourier Transform (FFT) method by adjusting the SM anomaly time series in the frequency domain (Fig. 3). This method allows for the identification of dominant frequencies and quantification of their respective contributions to the total variance, as represented by the Power Spectral Density (PSD) (Seo and Dirmeyer, 2022a). The FFT is applied to the preprocessed hourly SM time series for overlapped dates among the three datasets (in-situ observations and both reanalysis datasets based on LSMs). To ensure the continuity of hourly SM time series for the application of FFT, the preprocessed time series are concatenated. Both ERA5-Land and MERRA-2 datasets are used to adjust the SM diurnal cycle of in-situ measurements since the modelled time series are reliable with reference to their physically based simulation of SM dynamics (Fig. 2b). Based on the diurnal component of the multi-model averaged SM spectrum from both reanalyses, the harmonic variance of SM from in-situ observations is adjusted within only the 20- to 30-hour frequency band. Its mathematical formulation is followed as:

$$FFT_{obs}[freq_{30}:freq_{20}] = \frac{FFT_{obs}}{FFT_{LSMs}} \times FFT_{LSMs}[freq_{30}:freq_{20}], \qquad (1)$$



where $FFT_{obs}$ and $FFT_{LSMs}$ are the spectral power of SM time series from in-situ observations and averaged LSMs, respectively, and $freq$ is frequency domain in the PSD. The scaling factor ($FFT_{obs}/FFT_{LSMs}$), is applied to ensure continuity at the window boundaries, adjusting for difference in spectral characteristics between datasets (Seo and Dirmeyer, 2022a) (Fig. 3c). PSD also appears at negative frequencies, which do not physically exist, due to the symmetry property of the Fourier

Transform, where positive and negative frequency components mathematically mirror each other. The adjustment is consistently applied within the negative frequency range $[freq_{-20}: freq_{-30}]$. An updated hourly SM time series is then reconstructed using the Inverse FFT (IFFT) of the adjusted spectrum in the diurnal frequency range (Step 3 in Fig. 4d), after which the previously filtered low-frequency SM time series (Step 2; c.f., Fig. 4b) is added again (Step 4 in Fig. 4e).

Assuming SM measurements to be relatively insensitive to temperature during the nighttime hours (20:00–06:00 LST) due

to the absence of solar radiation, we further correct the mean of diurnal anomaly for each day. The nighttime mean of the adjusted SM anomaly is matched to the nighttime mean of the original SM anomaly (Step 5 in Fig. 4f), for each calendar day. This is achieved by subtracting the difference between the original and the adjusted nighttime means in the entire adjusted diurnal cycle. As this approach can result in negative values in the reconstructed SM time series for very dry soils, an additional adjustment is applied for those days to ensure physical plausibility. For calendar dates with negative SM values, the diurnal

amplitude is reduced using standard normal deviate scaling (SNDS; Koster et al. 2004; Seo et al. 2019; Seo and Dirmeyer, 2022a), thereby preventing negative SM values while preserving the daily mean SM. By applying the diurnal mean correction, most stations showed a decrease of SM by around 0–10%, indicating that the SM climatology in the observations has been overestimated due to sensor temperature sensitivity (Fig. 4g). This decrease is particularly prominent in the western US, where the SM is generally low and diurnal temperature range (DTR) is large. Hereafter, we refer to the adjusted ISMN SM based on

the LSM simulations as ISMN_adj.

### 3.3 Effect of diurnal temperatures on soil moisture-temperature coupling

We have analyzed how the temporal correlation between SM and TS varies with the climatology of the DTR ($T_{range} = T_{max} - T_{min}$). We examined the statistical relationship between the soil moisture-temperature coupling ($\rho = R(SM, TS)$) and $T_{range}$, across the observation sites, quantified as:

$$R(\rho, T_{range}) = \frac{E[(\rho - E[\rho])(T_{range} - E[T_{range}])]}{\sigma_\rho \sigma_{range}}, \tag{2}$$

where $E[\cdot]$ denotes the expectation operator, and $\sigma_\rho$ and $\sigma_{range}$ denote the standard deviation of $\rho$ and $T_{range}$, respectively. Since $T_{range}$ is defined as the difference between maximum temperature ($T_{max}$) and minimum temperature ($T_{min}$), the expression can be expanded as:

$$= \frac{1}{\sigma_\rho \sigma_{range}} \{E[(\rho - E[\rho])(T_{max} - E[T_{max}])] - E[(\rho - E[\rho])(T_{min} - E[T_{min}])]\}, \tag{3}$$

This can be rewritten using the definition of covariance ($Cov$) as:





$$= \frac{Cov(\rho, T_{max})}{\sigma_\rho \sigma_{range}} - \frac{Cov(\rho, T_{min})}{\sigma_\rho \sigma_{range}}, \tag{4}$$

To express each covariance term in the form of a correlation, we multiply the numerator and denominator of each term by

the corresponding standard deviations ($\sigma_{max}$ or $\sigma_{min}$):

$$= \frac{1}{\sigma_{range}} \{ \sigma_{max} \cdot R(\rho, T_{max}) - \sigma_{min} \cdot R(\rho, T_{min}) \}, \tag{5}$$

where $\sigma_{max}$ and $\sigma_{min}$ denote the standard deviation of $T_{max}$ and $T_{min}$, respectively. These equations show that the correlation between $\rho$ and $T_{range}$ can be decomposed into the separate contributions from $T_{max}$ and $T_{min}$. This study explores the correlation between $\rho$ and $T_{range}$ to understand the integrated effect of diurnal temperature variability on SM sensors.

### 3.4 Land coupling strength

To investigate land coupling between SM and LH, this study uses the Terrestrial Coupling Index (TCI; Dirmeyer, 2011; Seo and Dirmeyer, 2022b), which quantifies the influence of a source variable ($SV$) on a target variable ($TV$). TCI is a statistical metric that quantifies how strongly a target variable responds to variability in a source variable. It incorporates both the sensitivity between the two variables (e.g., coefficient correlation) and the magnitude of variability in the target (e.g. standard deviation). However, it does not imply causality and should be interpreted as a measure of statistical association only. When

LH is the target, positive or negative TCI values indicate that the L-A coupling chain is primarily triggered by SM or net radiation, respectively, corresponding to water- or energy-limited processes (see Fig. 2 in Seo et al., 2024). This metric is formulated as:

$$TCI(SV, TV) = r(SV, TV) \times \sigma(TV), \tag{6}$$

where $r$ represents the correlation coefficient between the time series of source and target variables, and $\sigma$ represents the standard deviation of target variable. In this study, the source and target variables are set as SM and LE, respectively, and TCI

is implemented with data sampled at multiple temporal scales (e.g., diurnal, daily, and monthly). Hourly time series are reconstructed by averaging the values from multiple days at each hour, after filtering them with a 24-hour centered moving average. Daily and monthly products are constructed by averaging the original time series. Although the relationship between SM and LH is not exactly linear, the linear dependencies dominate over much of the globe (see Fig. 4 in Hsu and Dirmeyer, 2021), supporting the applicability of the TCI.

### 4 Results

### 4.1 Evaluation of SM diurnal cycle and its relationship with temperature

To determine whether the FFT-based adjustment successfully corrects the SM diurnal cycle, we compare the diurnally adjusted time series against reference sensors (non-electrical or high-frequency dielectric-based sensors), which are less affected by temperature-driven biases. Since the reference sensors and dielectric-based sensors are not co-located, the closest





sensor pairs within 200km are used for comparison, resulting in a total of 20 pairs (Fig. 5). The minimum of SM diurnal cycle is expected in the afternoon because the peak in incoming solar radiation enhances ET during the daytime. However, the diurnal cycle in the ISMN shows a peak in the afternoon, contradicting the realistic SM behavior based on the water budget balance. The diurnal time series in ISMN$_{adj}$ show a peak in the morning and a minimum in the afternoon, aligning well with the physically reliable feature and showing coherent SM behavior relative to the reference sensors which exhibit their minimum

values in the afternoon (Fig. 5a). There is a large spread among diurnal time series within ISMN due to the diversity in climate zones and land cover types across the sites. The correlation of the adjusted SM diurnal cycle is increased by 0.6, compared to the result from the original time series, indicating that the adjusted time series better captures the expected diurnal behavior (Fig. 5b). The spatial distribution of the relationship between surface SM and TS in ISMN unrealistically exhibits a positive correlation between these two variables over relatively arid regions, showing an SM peak in the afternoon (Figs. 6a, d). In

contrast, the results from the combined LSMs generally indicate a negative correlation, characterized by a morning SM peak (Figs. 6b, e). When the in-situ observations are diurnally adjusted using the LSMs, the regions with spuriously positive correlations in the original time series shift to physically consistent negative correlations (Figs. 6c, f).

To examine how soil moisture–temperature coupling is influenced by temperature sensitivity during both daytime and nighttime, we classified the diurnal correlation between SM and TS ($\rho = R(SM, TS)$) based on the $T_{range}$. As described in

Section 3.3, by expressing $T_{range}$ in terms of $T_{max}$ and $T_{min}$, we can separately quantify the effects of daytime temperatures ($R(\rho, T_{max})$) and nighttime temperatures ($R(\rho, T_{min})$) on $\rho$. Since the Maxwell-Wagner effect primarily arises with high temperature, $R(\rho, T_{max})$ shows a larger positive value compared to $R(\rho, T_{min})$ (Table 1). $R(\rho, T_{range})$ exhibits a larger value than $R(\rho, T_{max})$, primarily because $\sigma_{range}$ is smaller than $\sigma_{max}$ (cf., Eq. 5). Therefore, in regions characterized by a large $T_{range}$, $\rho$ tends to exhibit positive values (Fig. 7a). A similar pattern is also observed in regions where the SM climatology is

relatively dry (Fig. S2). On clear days, incoming radiation at the surface enhances LH, leading to soil drying, which subsequently increases the partitioning toward SH, ultimately raising daytime temperatures. This classification further enables examination the effect of the FFT-based adjustment on SM measurements across various $T_{range}$ regimes (Fig. 7b). When $T_{range}$ is small, the original ISMN data exhibits a concentration of diurnal correlations on the negative side, shifting toward positive values as $T_{range}$ becomes large. Such spurious SM–temperature coupling is significantly mitigated in the ISMN$_{adj}$

data, particularly in regions characterized by large $T_{range}$ and arid SM conditions.

We additionally assess the effect of the diurnally adjusted SM time series on SM–temperature coupling according to the Köppen-Geiger climate classification (Fig. S3), aggregated into four first-level climate groups: Tropical, Temperate, Continental, and Dry (Fig. 8a). The Polar category is excluded from the analysis due to its limited sample size (n=5) and spatially concentrated over the Tibetan Plateau (Fig. S3) rather than at high-latitude polar regions. The correlation between

SM and TS, quantifying SM–temperature coupling, is predominantly positive in the original ISMN data, particularly in Tropical and Dry regions. In Tropical regions, both dry and wet subtypes exhibit positive diurnal correlations (not shown), likely attributable to the consistently high temperatures ($T_{min} \geq 18°C$) by the definition of the Köppen-Geiger classification.



Dry regions are characterized by high $T_{range}$, typical of arid environments as discussed above, resulting from high $T_{max}$. The other climate classifications—Temperate and Continental—exhibit bimodal distributions with both positive and negative correlations (Fig. 8a), prompting further subdivision based on second-level Köppen-Geiger types. For instance, subtypes "s" (dry summer) and "w" (dry winter) are grouped as dry, while "f" (fully humid) and "m" (monsoonal) are grouped as wet. This subdivision clarifies the diurnal correlation patterns, revealing notably spurious positive correlations in dry climates (Figs. 8b, c). The diurnally adjusted SM observations exhibit negative correlations across all climate classifications, consistent with both reanalysis products, although ERA5-Land shows weaker negative SM–temperature coupling compared to MERRA-2. The difference in $R(SM, TS)$ between ERA5-Land and MERRA-2 is primarily due to inconsistencies in their LH. MERRA-2 shows an earlier peak in LH and SH than ERA5-Land. This leads to a delayed onset of surface cooling based on energy balance (Fig. S4) and subsequently results in a later peak of TS. Consequently, MERRA-2 shows a more pronounced out-of-phase relationship between SM decrease and TS increase, which results in a stronger negative correlation in MERRA-2 than in ERA5-Land.

### 4.2 Land segment-based evaluation of L-A interaction

While the previous analysis primarily focuses on the relationship between SM and TS, we further turn to flux tower observations to understand the influence of the SM adjustment on L-A interactions with LH, using data from the local warm season (MJJAS: May-September for the Northern Hemisphere; NDJFM: November-March for the Southern Hemisphere). We employ the TCI metric, which quantifies the statistical influence of SM on LH by multiplying their correlation coefficient by the variability of LH, enabling its implementation across various temporal scales (i.e., monthly, daily, and hourly) (Fig. 9). At monthly and daily timescales, $r(SM, LH)$ tend to be negative, resulting in negative TCI values associated with these negative correlations, as most flux tower sites are located in energy-limited regions. The coupling strength at the monthly scale is stronger than at the daily timescale, despite a larger standard deviation in daily LH, because the monthly time series have pronounced seasonal cycles.

For the L-A interactions at the sub-daily time scale, the increased solar radiation during daytime enhances ET, which is characteristic of energy-limited coupling, necessarily leading to a daytime decrease in SM and thus resulting in a negative $r(SM, LH)$. However, the data from ISMN typically shows a positive $r(SM, LH)$, as discussed previously. In contrast, ISMN_adj successfully captures the energy-limited coupling, indicated by negative TCI values, and exhibits the strongest coupling strength across all temporal scales. The diurnal variation of LH driven by solar radiation is most prominent at sub-daily scales and progressively smooths out when averaged over longer timescales (Fig. 9c). This suggests that analyzing L-A coupling at sub-daily timescales using in-situ SM data without taking into account the physically inconsistent diurnal behavior of dielectric SM measurements would impede an accurate understanding of L-A interactions (Fig. 9a and 9b).





**5 Summary and Conclusions**

This study has introduced a Fourier transform-based approach specifically designed to correct temperature-induced errors prevalent in SM in-situ measurements obtained from dielectric-based sensors. These temperature-induced inaccuracies arise primarily from the sensors' heightened sensitivity associated with the Maxwell–Wagner polarization effect, which significantly affects the dielectric properties of the soil as temperature increases. These errors lead to physically unrealistic diurnal cycles

characterized by spurious afternoon peaks in SM, particularly near the surface where the diurnal cycle of temperature is greatest. Afternoon is precisely when evapotranspiration is highest, and soil physics suggests SM should be at its minimum under precipitation-free conditions.

To ameliorate such temperature-induced errors, the proposed adjustment method leverages physically consistent diurnal SM time series derived from two reanalysis datasets: ERA5-Land and MERRA-2. These models provide robust, physically

constrained representations of SM behavior based on water balance closure, making them suitable benchmarks of the diurnal cycle for correcting the dielectric-based measurements. Specifically, the method adjusts the spectral power of the observed SM cycles around daily timescales, aligning them more closely with reliable hourly SM time series. Additionally, a diurnal mean correction (20:00–06:00 LST) is incorporated, assuming that the sensor temperature sensitivity is limited during nighttime periods when temperature-induced errors are minimal, thereby serving as a baseline for the correction. Notably, it

underscores that the SM climatology in the observations can have wet bias due to sensor temperature sensitivity.

Validation against reference sensors, whose design is known to have less temperature sensitivity, demonstrates significant improvements. Adjusted SM data (ISMN$_{adj}$) effectively transitions from exhibiting unrealistic afternoon peaks to displaying physically reasonable diurnal cycles characterized by morning peaks and subsequent afternoon minima, closely mirroring the soil moisture dynamics dictated by evapotranspiration processes (Fig. 5). The skill improvement in ISMN$_{adj}$, measured by

temporal correlation of SM hourly time series with the reference observations, is statistically significant, having $\Delta R \sim 0.6$, compared to the ISMN raw product. The Fourier-based adjustment successfully mitigates these spurious positive correlations between SM and temperature, converting them to physically consistent negative correlations, reflecting the true interactions between SM and temperature linked by evapotranspiration dynamics. Moreover, the impact of the diurnal adjustment is examined within the Köppen-Geiger climate classification, particularly indicating the efficacy in arid regions characterized by

pronounced temperature fluctuations with large DTR.

To further validate the effect of the SM diurnal adjustment on characterizing L-A interactions at sub-daily scales, this study demonstrates the improvement using flux tower observations. The Terrestrial Coupling Index (TCI), which quantifies the statistical relationship between SM and LH, is employed across multiple temporal scales (monthly, daily, and hourly). At monthly and daily timescales, $r(SM, LH)$ is negative due to most of flux sites located over mid-latitude regions, in which

increased LH leads to a decrease in SM (energy-limited coupling). However, at sub-daily timescale, original ISMN, uncorrected SM data erroneously show positive correlations, inconsistent with the physically balanced water budget. After



applying the diurnal phase adjustment, hourly SM data of ISMN$_{adj}$ accurately results in negative correlations, thereby facilitating an accurate understanding of true L-A coupling at sub-daily scales.

In conclusion, the Fourier transform-based correction method at sub-daily timescales substantially enhances the realism and reliability of SM diurnal cycle in the observations. By rectifying temperature-induced sensor errors, this generalizable approach significantly improves the reality of SM behavior in observational data. This improvement enhances the reliability of in-situ observations providing a robust foundation for comprehensively understanding SM dynamics and L–A interactions at sub-daily timescales, an area previously hindered by observational limitations, thereby benefiting model improvement, satellite validation, and climate monitoring.

**Acknowledgements**

This research was conducted within Global - Learning & Academic research institution for Master's·PhD students, and Postdocs (LAMP) Program of the National Research Foundation of Korea (NRF) grant funded by the Ministry of Education (RS-2023-00301702). Eunkyo Seo was supported by the National Research Foundation of Korea (NRF) grant funded by the Korea government (MSIT) (RS-2025-02363044).

**Code availability**

The source code used in the diurnal filtering is shared on the Github (https://github.com/wnssud0621/Diurnal-filtering)

**Data availability**

In-situ observations from ISMN can be accessed and downloaded publicly via their website at https://ismn.earth/en/data/data-download/. Soil moisture and LH from flux tower observations can be accessed from the following websites: FLUXNET2015 at https://fluxnet.org/data/fluxnet2015-dataset/, AmeriFlux at https://ameriflux.lbl.gov/data/download-data/, and ICOS at https://www.icos-cp.eu/data-products. Hourly data from the Copernicus Climate Change Service (C3S) ERA5-Land reanalysis are freely available through its online portal at https://cds.climate.copernicus.eu/datasets/reanalysis-era5-land?tab=overview. The hourly MERRA-2 data are available for free through the NASA Goddard Earth Sciences (GES) Data and Information Service Center (DISC) at https://disc.gsfc.nasa.gov/datasets?project=MERRA-2.

**Author contributions**

Conceptualization: JH, ES, and PAD. Investigation, methodology, and formal analysis: JH and ES. Writing (original draft preparation): JH. Writing (review and editing): ES and PAD.





**Competing interests**

The contact author has declared that neither of the authors has any competing interests.



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





**Figure**

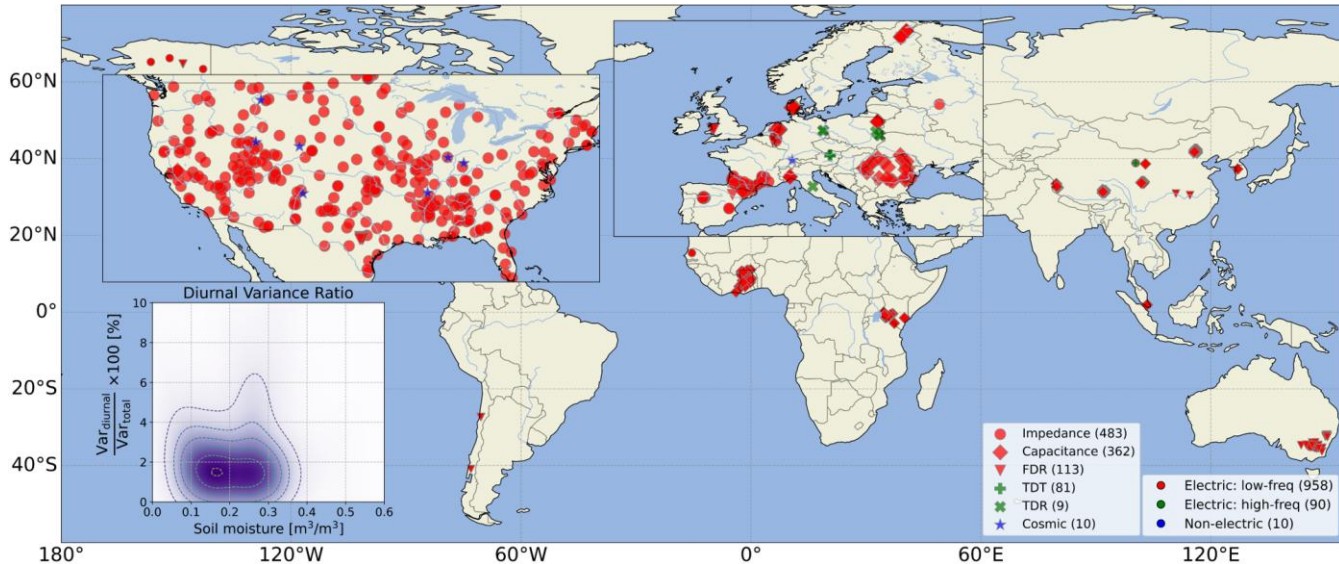

**Figure 1: Locations of in-situ observation sites of SM (N=1058) from the International SM Network (ISMN). Red and blue dots indicate sensors that use electrical properties (dielectric constant) to observe SM and do not use that method, respectively. Sensors that operate at high frequences, insensitive to temperature biases, are shaded green. The lower-left figure shows the ratio of diurnal**
**variance against total variance in the SM measurements. The bracketed numbers in the lower-right legend denote the number of sites adopting each corresponding sensor.**



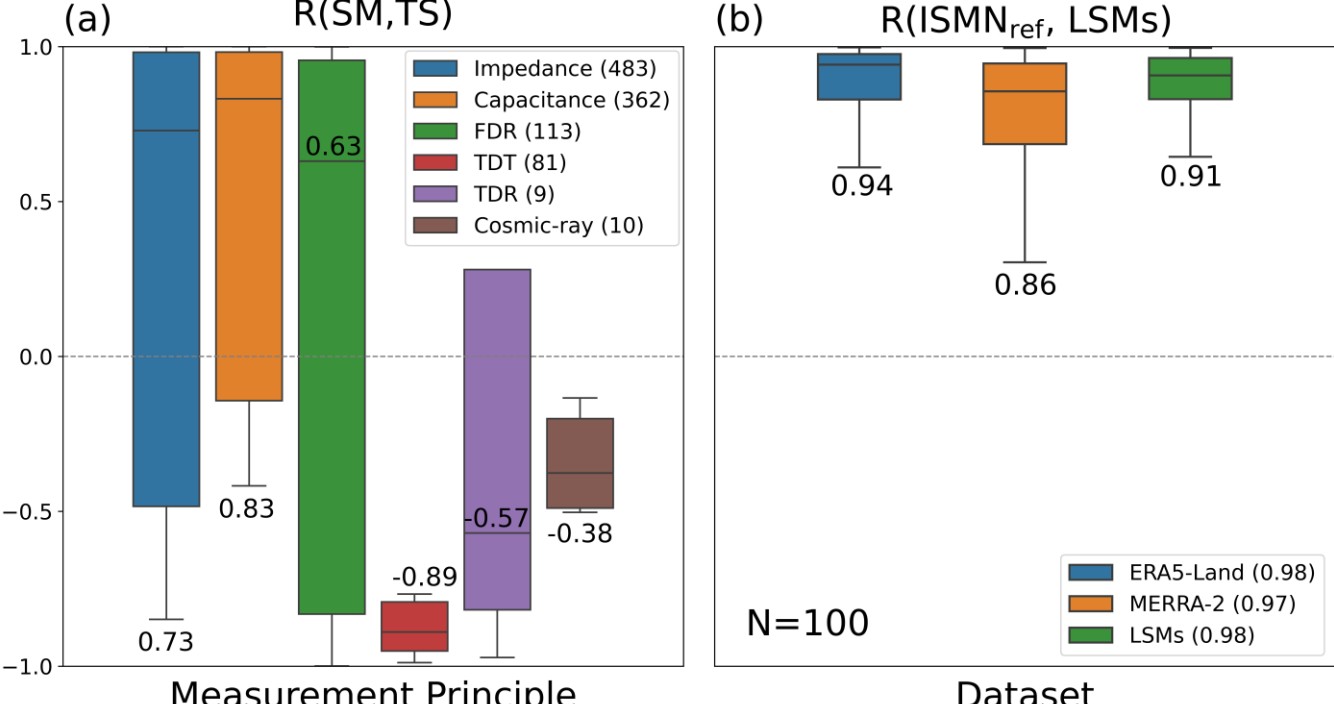

**Figure 2: Boxplot of the temporal correlation within the diurnal cycle of (a) hourly time series of surface SM and soil temperature for each sensor used to measure SM and (b) modelled surface SM against reference sensors (those with a negative correlation to diurnal temperature: TDT, TDR, and cosmic-ray), where median values are denoted beside each boxplot. Values in parentheses in the legend represent (a) the number of stations using each sensor type for measuring SM, and (b) the correlation between the 24-hour diurnal cycles of reference and modelled SM, based on 24-hour cycles constructed by median values across 100 stations at each hour. LSMs (green) indicate the averaged SM time series from ERA5-Land (blue) and MERRA-2 (orange).**

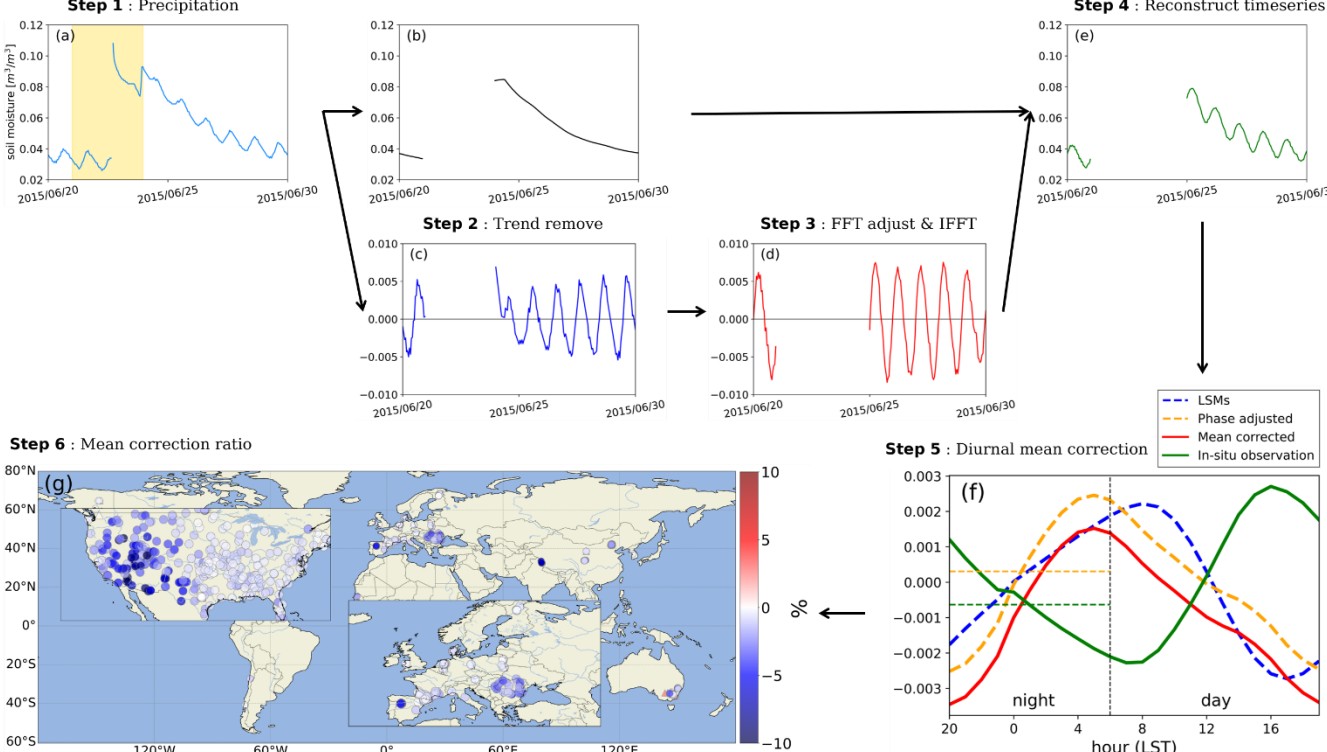

**Figure 3: Data preprocessing steps for example station REMEDHUS- ElTomillar (41.35°N, 5.49°W): (a) Removal of days with precipitation events, with yellow shading indicating precipitation dates. (b) Running mean calculated using a 24-hour centered moving average. (c) SM anomaly derived by subtracting (b). (d) Adjusted SM anomaly using the Fourier Transform method. (e) Reconstruction of time series obtained by adding (d) and (b). (f) Diurnal time series showing the original in-situ observation (green), model-based data (blue dashed line), phase adjusted data (yellow dashed line), and the final adjusted result (red line). (g) Mean corrected ratio over the global map, with negative percentages in blue and positive percentages in red.**





**Figure 4: Power Spectral Density (PSD) analysis of surface SM from (a) in-situ measurement, (b) reanalysis datasets (red: ERA5-Land, and green: MERRA-2), and (c) diurnally adjusted in-situ measurement at REMEDHUS-ElTomillar station (41.35°N, 5.49°W).** The gray shaded area in each panel represents the adjusted frequency windows and the black dotted vertical lines indicate the 20- and 30-hour variance domains. The yellow line in (c) represents the original power of surface SM in the diurnal time window.





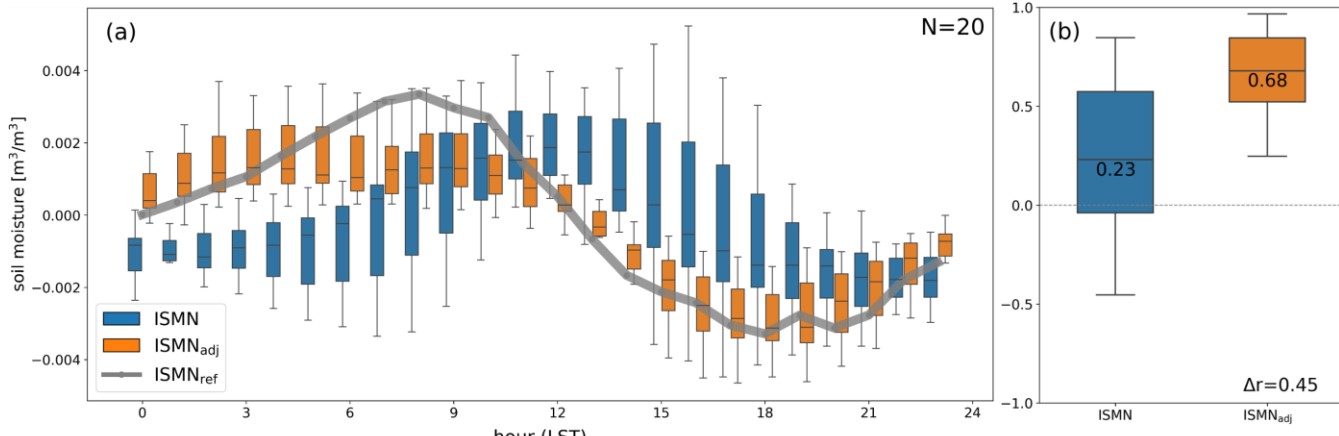

**Figure 5: (a) Diurnal cycle of surface SM, filtered by a 24-hour running mean from electrical-based ISMN (blue), ISMN$_{adj}$**
**(orange boxplot), and reference sensors (gray line). To compare the impact of diurnal adjustment on electric-based measurements with the reference sensors, the 20 closest electric-based observations are sampled within 200 km from the reference sensors. (b) Boxplot of the temporal correlation coefficient of diurnal time series of surface SM from electrical-based ISMN (blue) and ISMN$_{adj}$ (orange) against the corresponding reference observations.**







**Figure 6: Spatial distribution of the temporal correlation coefficient of hourly time series of surface soil moisture (SM) and temperature (TS) from (a) ISMN, (b) the mean of land reanalysis datasets (LSMs: ERA5-Land and MERRA-2), and (c) ISMN$_{adj}$, where red triangles and blue circles indicate positive and negative correlation coefficient, respectively. The diurnal maximum phase of surface SM in (d) ISMN, (e) LSMs, and (f) ISMN$_{adj}$.**





**Figure 7: (a) Spatial distribution of correlation coefficients between diurnal cycles of SM and TS from ISMN, classified by $T_{range}$, where upward- and downward-pointing triangles indicate positive and negative correlations, respectively. The lower-left scatter plot shows the relationship between climatological DTR and SM, where colors correspond to the same DTR range used in the map. The spatial correlation coefficient along with its corresponding p-value is also shown. (b) Violin plots of the probability distribution of correlation coefficients (on Y axis) of ISMN (blue) and $ISMN_{adj}$ (orange), in the same three categories according to $T_{range}$. The bracketed numbers below the x-axis represent the number of stations classified in each DTR category. The total number of stations is decreased to 745, compared with Fig. 1, because only sites with concurrent SM and TS records for at least 92 days are used to calculate the correlation coefficient between variables.**



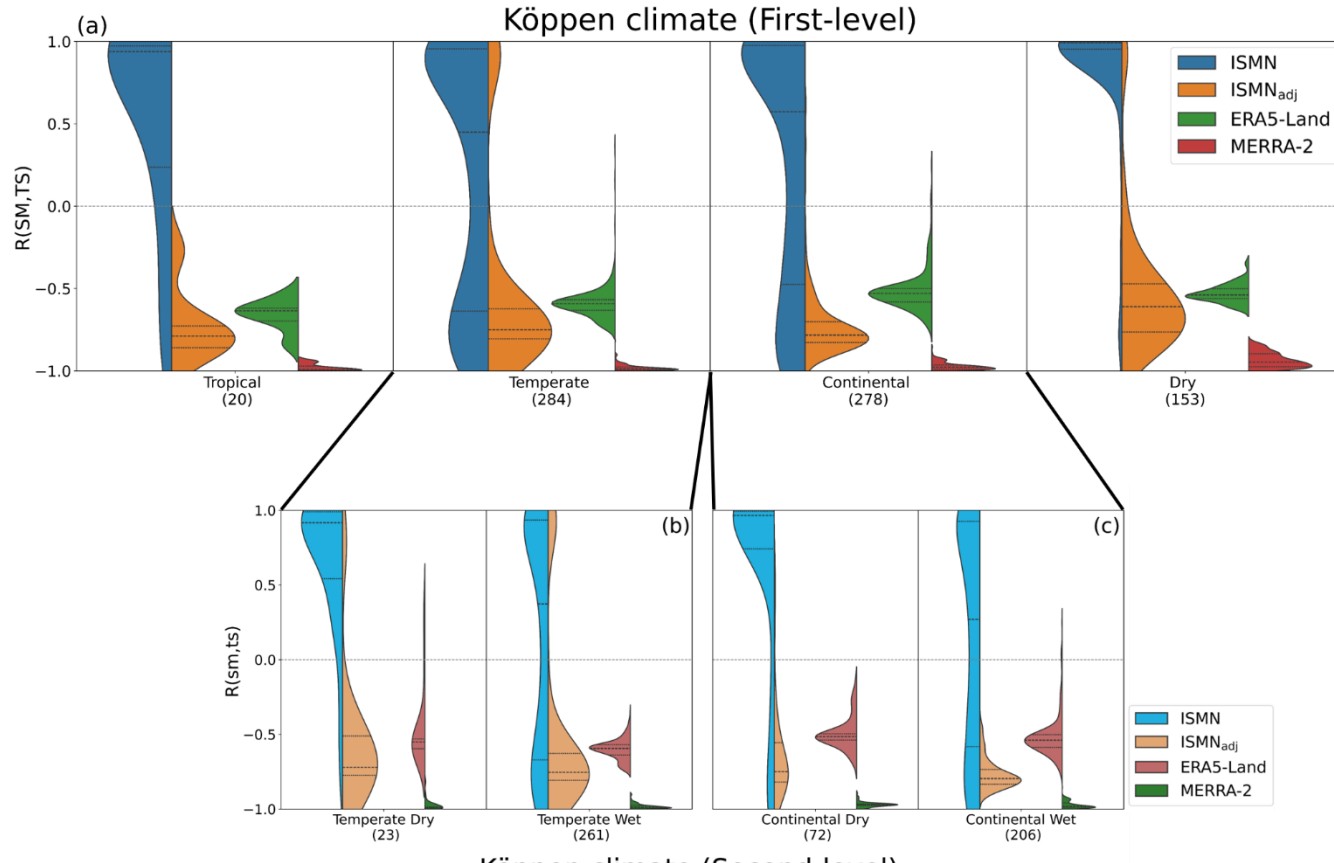

**Figure 8: Same as in Fig. 7b, but grouped by (a) the first-level Köppen-Geiger classification, and the second-level classification for (b) Temperate and (b) Continental climate zones. The classification in ERA5-Land (green) and MERRA-2 (red) is also represented, corresponding to the in-situ observations. The Polar (N=5) and Unknown (N=5) climate types are excluded in this 645 analysis, so 735 stations are used in this classification.**



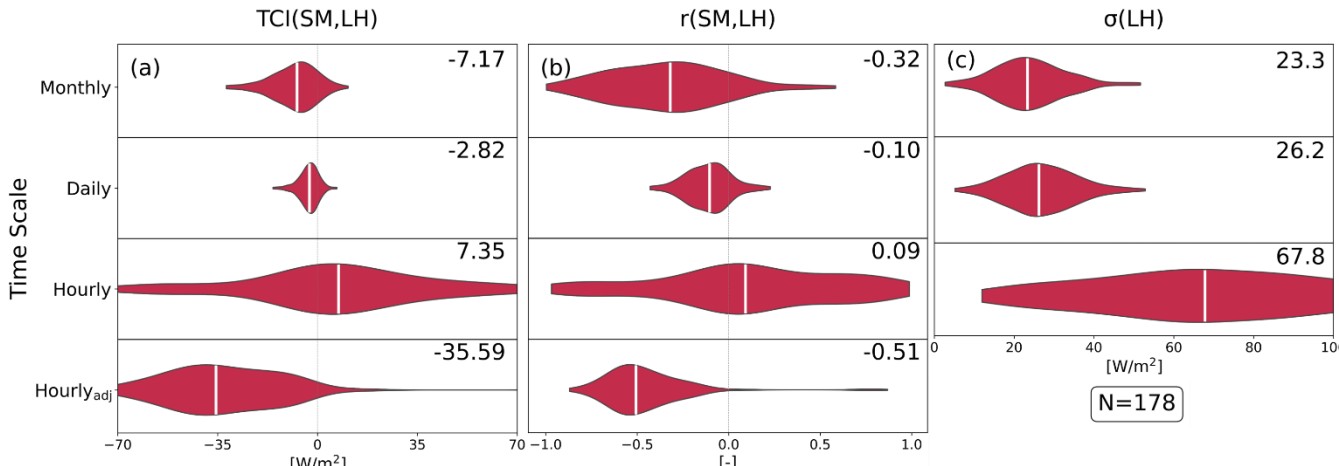

Figure 9: Violin plot of land coupling and its components from flux tower observations (N=178) in monthly (top row), daily (second row), and hourly (third row) and diurnally adjusted hourly (bottom row) timescale during the hemispheric warm season (Northern and Southern Hemisphere MJJAS and NDJFM, respectively). Since the diurnal adjustment is conducted only on the SM time series, the standard deviation term remains identical to the diurnally unadjusted result. (a) TCI(SM, LH) is used to measure the strength of the land coupling, which is a term multiplied by (b) the correlation coefficient between SM and LH and (c) the standard deviation of LH. The values in the upper-right corner of each panel indicate the median value.





|  | $R(\rho,T)$ | $\sigma(T)$ |
|---|---|---|
| $T_{max}$ | 0.497 | 6.0 K |
| $T_{min}$ | 0.081 | 4.2 K |
| $T_{range}$ | 0.554 | 4.8 K |

**Table 1: The spatial correlation coefficient between $\rho$ and climatological $T_{max}, T_{min}$, and $T_{range}$ across 983 observation sites (left column), along with their standard deviations (right column).**

660