# Peer review of "Adjusting Diurnal Error in In-Situ Soil Moisture Measurements via Fourier Time-Filtering Using Land Surface Model Datasets"

_EGUsphere, 2025_

## Referee Comment (RC2)

Comments on the *"Adjusting Diurnal Error in In-Situ Soil Moisture Measurements via Fourier Time-Filtering Using Land Surface Model Datasets"* by J. Han et al.

The manuscript presents a Fourier transform–based approach to correct temperature-induced diurnal artifacts in in-situ soil moisture (SM) measurements from dielectric sensors. Constrained by the physically consistent land surface model (LSM) reanalysis products (ERA5-Land and MERRA-2), the authors aim to restore physically realistic diurnal behavior of dielectric-based SM and improve the interpretation of land–atmosphere interactions at sub-daily timescale.

The topic is highly relevant to the hydrology and land–atmosphere coupling communities. The manuscript is generally well written, methodologically innovative, and supported by extensive datasets. The proposed correction has the potential to substantially improve the usability of high-frequency in-situ SM observations. However, several issues need to be addressed before the manuscript can be considered for publication.

Major comments:

1. Introduction: Literature review on correction of SM is too limited, only the second last paragraph of the introduction section. At least correction methods of the same type as those used in this study should be reviewed in detail.
2. Data: Three land reanalysis datasets (ERA5-Land, MERRA-2, GLDAS) are discussed, but only two are used in the adjustment. The rationale for this choice is not sufficiently clear.
3. Methodology: The manuscript excludes all days identified as "rainy" based on SM tendency thresholds or precipitation totals. This approach is questionable, particularly in arid and semi-arid regions. Light precipitation events often evaporate rapidly and may have negligible impacts on daily soil moisture, especially near the surface.
4. Methodology: The power spectral densities shown in the manuscript are unevenly distributed in frequency space, with sparse low-frequency bins and dense high-frequency bins. This makes interpretation of the diurnal peak less robust. Recommend applying frequency or logarithmic smoothing to the power spectra.
5. Results: The evaluation strategy compares adjusted SM time series with reference sensors located up to 200 km away. Given the extreme spatial heterogeneity of

soil moisture, such comparisons raise concerns about physical meaning. Even at distances of a few meters or hundred meters, SM can differ substantially due to soil texture, vegetation, topography, and land management.

Minor points:

1. The correction is only for dielectric-based SM, this should be pointed out in the Title.
2. L124-125: This has been reported in the introduction and do not need to repeat here.
3. L168: References introducing eddy-covariance method are needed here, there are multiple classical papers and books. Pastorello et al., 2020 is not a perfect one.
4. L172: Reference for FLUXNET2015 is missing here. Pastorello et al., 2020 is the right citation.
5. L205: Fig4a -> Fig 3a? And similar issues below.
6. L239: What adjustment?